# GREEN PRUNING: LAYER INTERDEPENDENCE-AWARE CNN PRUNING FOR RESOURCE EFFICIENCY

## ABSTRACT

The rising computational demands of pruning algorithms have heightened challenges about their energy consumption and carbon footprint in convolutional neural networks. We address these challenges from two perspectives. First, we introduce new evaluation metrics for pruning: a Resource Efficiency (RE) metric, which quantifies the computational cost required to achieve a target accuracy, and a system-agnostic framework for assessing the relative carbon efficiency of pruning algorithms. Together, these metrics enable fair and consistent comparisons of pruning methods with respect to both efficiency and sustainability. Second, we present a **green pruning technique**, a data-free method that explicitly models inter-layer dependencies to provide a more reliable filter selection criterion. To further minimize computational overhead, our approach incorporates a low-complexity oblivious algorithm that leverages weak submodularity, ensuring efficiency without requiring iterative dataset passes.

## 1 INTRODUCTION

Pruning Convolutional Neural Networks (CNNs) has emerged as a prominent approach to model compression, offering the potential to eliminate redundant filters and enable efficient deployment. However, despite its promise, pruning faces two critical challenges: environmental cost and practical feasibility on edge devices. First, while pruning is often framed as a strategy for improving efficiency, the pruning process itself can be computationally intensive (Cheng et al., 2017). Sophisticated, data-driven pruning methods typically require repeated forward and backward passes over large datasets, incurring significant energy consumption. As models and datasets continue to scale, the associated carbon footprint of pruning methods has become an urgent challenge. Addressing this issue is not optional but essential, underscoring the critical importance of advancing **Green Pruning** as a core direction for sustainable AI. In large-scale environments such as cloud platforms or GPU clusters, this environmental cost compounds, raising questions about the true sustainability of current pruning practices. Consequently, lightweight, data-free methods—such as magnitude-based pruning—remain attractive because they substantially reduce computational overhead, power consumption, and ultimately environmental impact, even if their accuracy is less competitive. Second, edge devices introduce an additional layer of constraint. With limited memory and processing capabilities, they cannot accommodate pruning methods that involve expensive iterative computations, large dataset passes, or complex heuristics. For these scenarios, pruning must not only reduce inference costs but also be inherently efficient and easy to implement during the compression process itself. This highlights the need for resource-efficient pruning strategies that balance accuracy, computational simplicity, and sustainability.

Current evaluations of pruning methods largely hinge on two metrics: accuracy degradation and reductions in FLOPs and parameters of the pruned network (Yvinec et al., 2023), (Luo et al., 2018), (Lin et al., 2020). While informative, these measures present a narrow view, ignoring critical dimensions such as resource efficiency and environmental sustainability. As AI systems scale and deployment on edge devices becomes increasingly common, continuing to overlook these factors risks making pruning methods less relevant in practice. There is a clear need for evaluation frameworks that move beyond accuracy alone, incorporating efficiency and sustainability as objectives.

In this paper, we introduce, for the first time, metrics to evaluate pruning methods with respect to both resource efficiency and sustainability. Specifically, we propose a **Resource Efficiency (RE)**

metric that quantifies the computational cost required to achieve a target level of accuracy. In addition, we analyze the carbon footprint of pruning methods. While absolute emissions depend on the underlying hardware and system configuration, we present a system-agnostic **Relative Carbon Efficiency (RCE)** metric that enables fair and consistent evaluation of sustainability across different approaches. Beyond these metrics, we propose a novel CNN compression method. Our approach explicitly models inter-layer dependencies to provide a more reliable pruning criterion than conventional data-free techniques. Unlike magnitude-based methods such as L1 pruning (;and Hans Peter Graf, 2017), which evaluate filters in isolation, our method leverages information from adjacent layer parameters, yielding a more holistic and effective pruning mechanism. To further reduce the computational burden commonly associated with greedy pruning algorithms, our method employs a low-complexity *oblivious algorithm*, exploiting the $\gamma$-weakly submodular property of the underlying optimization problem. This ensures that filter selection is both theoretically grounded and highly efficient, making our method particularly well-suited for deployment on resource-constrained edge devices. Importantly, our proposed method advances not only computational efficiency but also environmental sustainability. By reducing the number of operations required for filter selection by several orders of magnitude compared to existing methods, it substantially lowers energy consumption and the associated **carbon footprint**, while preserving the simplicity and robustness of norm-based approaches. Our results demonstrate that our method is not only a resource-efficient and sustainable pruning method but also achieves accuracy on par with state-of-the-art techniques.

This paper makes two primary contributions. First, we introduce novel evaluation metrics for pruning, including a Resource Efficiency metric and a system-agnostic framework for measuring the carbon footprint, enabling fair comparison of pruning methods with respect to both efficiency and sustainability. Second, we propose a new pruning method that explicitly models inter-layer dependencies and employs a low-complexity oblivious algorithm. our method achieves high resource efficiency and significantly reduces the carbon footprint of pruning, while maintaining accuracy comparable to state-of-the-art methods.

## 2 RELATED WORK

Pruning techniques are commonly divided into two categories: *structured* pruning (Lin et al., 2020; ;and Hans Peter Graf, 2017; Luo et al., 2018; Alwani et al., 2021; Qian et al., 2023; Lin et al., 2021) and *unstructured* pruning (Ding et al., 2019; Frankle and Carbin, 2019; Han et al., 2015). Structured pruning is generally preferred in practice, as the resulting models retain compatibility with standard hardware and yield more predictable performance improvements. Several factors influence the complexity and efficiency of pruning methods. Broadly, a pruning method is defined by two primary components: (i) the parameters to be pruned, such as filters in the convolutional layers of CNNs, and (ii) the dataset employed during the pruning process (Cheng et al., 2017).

Several factors impact the complexity and efficiency of pruning methods including A pruning methods is primarily characterized by two key components: the parameters (e.g., filters) in the convolutional layers of CNNs and the input dataset used by the network (Cheng et al., 2017). Pruning methods can also be distinguished by their reliance on dataset information. *Data-based* approaches evaluate convolutional outputs using training samples, either within the *current layer* or in *adjacent layers*, with the latter offering more comprehensive insights but at higher computational cost (El Halabi et al., 2022; Luo et al., 2018; Tofigh et al., 2022; Lin et al., 2020; Qian et al., 2023; Lin et al., 2019; He and Xiao, 2024). In contrast, *data-free* methods select filters solely from convolutional parameters, typically via *filter norm* or *filter correlation* criteria (;and Hans Peter Graf, 2017; He et al., 2019; Yvinec et al., 2021; 2023; He and Xiao, 2024). While data-based methods generally achieve superior accuracy, they are computationally expensive; data-free methods are more efficient but often less effective. Despite these variations, most pruning strategies remain heuristic, lack theoretical guarantees, and pose computational challenges that hinder practical deployment, particularly on resource-constrained edge devices (Luo et al., 2018; Lin et al., 2020; ;and Hans Peter Graf, 2017; He et al., 2019).

## 3 GREEN PRUNING METRICS

Green pruning, as a sustainable and environmentally conscious approach, requires accounting for the carbon footprint incurred during the filter selection process. Since this footprint inherently depends on the hardware and system used, a system-dependent evaluation is problematic for two reasons. First, most pruning methods lack sufficient system-level information to allow accurate estimation of carbon emissions. Second, a fair comparison across methods would require re-implementing them on the same system, which is often impractical. These limitations underscore the need for a system-agnostic framework to enable consistent and fair comparison of the carbon footprint across pruning methods.

### 3.1 RELATIVE CARBON EFFICIENCY (RCE)

Estimating the $CO_2$ emissions of a pruning method requires considering both its algorithmic complexity and the underlying hardware. To enable fair comparison, we introduce the **Relative Carbon Efficiency (RCE)**, defined as the ratio of the carbon footprint of two pruning methods:

$$\text{RCE} = \frac{RC(P_1)}{RC(P_2)}, \tag{1}$$

where $P_1$ and $P_2$ denote two pruning methods under comparison. We adopt the estimation methodology from (Lannelongue et al., 2021), where the carbon footprint is computed as:

$$\text{Carbon Footprint} = \text{Energy Needed} \times \text{Carbon Intensity}, \tag{2}$$

with energy consumption estimated as:

$$\text{Energy Needed} = \text{Runtime} \times (\text{CPU Power Draw} \times \text{Usage} + \text{Memory Power Draw}) \tag{3}$$
$$\times \text{PUE} \times \text{Multiplicative Factor}.$$

Among these terms, the only algorithm-dependent factor is the *runtime*. To design a system-agnostic framework, and in the absence of precise runtime measurements for all methods, it is standard practice to use the number of floating-point operations (FLOPs) as a hardware-independent proxy for runtime, i.e.,

$$\text{Runtime} = \frac{\mathcal{N}}{\text{FLOP/s}}, \tag{4}$$

where $\mathcal{N}$ denotes the number of operations required by the pruning method to remove a given number of filters, and FLOP/s represents the constant compute capability of the system. Substituting this into equation 1, RCE can be reformulated as:

$$\text{RCE} = \frac{\mathcal{N}_1}{\mathcal{N}_2}, \tag{5}$$

where $\mathcal{N}_1$ and $\mathcal{N}_2$ denote the number of operations required by pruning methods $P_1$ and $P_2$, respectively, to prune the same number of filters. According to equation equation 5, the RCE between two pruning methods depends only to the number of the operations required by each pruning method. This methodology is supported by prior studies identifying FLOPs as a dominant driver of energy consumption under standardized conditions (Patterson et al., 2021; Schwartz et al., 2020).This metric serves as a practical and meaningful proxy for estimating the relative environmental impact of each pruning technique.

### 3.2 RESOURCE EFFICIENCY (RE)

While maintaining the accuracy of the pruned network is crucial, a pruning method needs also be practical and suitable for deployment on devices. In other words, it must be low-complexity and fast, requiring minimal device resources during the pruning process. To address this, we propose a metric for evaluating the resource efficiency (RE) of a pruning method, which considers both accuracy and the computational complexity of the pruning process. This metric is defined as:

$$RE = \left( \frac{\text{ACC}}{\text{MAC}} \right), \tag{6}$$

where ACC represents the accuracy of the pruned network and MAC denotes the number of multiply-accumulate operations required in the pruning process. This metric provides a balanced evaluation of pruning methods, highlighting the trade-off between preserving accuracy and reducing computational costs, thereby offering a comprehensive measure of a method's overall efficiency.

## 4 PROPOSED GREEN PRUNING METHOD

In this section, we introduce a pruning strategy aimed at achieving resource efficiency and sustainability by maximizing the accuracy of the pruned network while ensuring that the pruning process itself remains computationally simple.

Pruning $k$ filters from the $l^{\text{th}}$ convolutional layer ($l = 1, \ldots, \mathbb{L}$) out of a total of $H_l$ filters is equivalent to retaining the $H_l - k$ most important filters. Formally, selecting the subset of $H_l - k$ filters with the highest importance values can be expressed as the following optimization problem:

$$\max_{\substack{S \subset \{1, \ldots, H_l\} \\ |S| = H_l - k}} \mathcal{L}(S), \tag{7}$$

where $\mathcal{L}(S)$ denotes the importance of the filter subset indexed by $S$. Solving this problem is NP-hard and thus computationally intractable for large-scale networks.

We define the importance function $\hat{\mathcal{L}}$ over a subset of filter indices $S \subset 1, \ldots, H_l$ as:

$$\hat{\mathcal{L}}(S) := \max\{|F^l[:,:,:,S]|_2^2, ; |F^{l+1}[:,:,S,:]|_2^2\}, \tag{8}$$

where $F^l[:,:,:,S]$ denotes the tensor formed by the filters in the $l^{\text{th}}$ layer with indices in $S$, and $F^{l+1}[:,:,S,:]$ denotes the tensor corresponding to the channels of the filters in the $(l+1)^{\text{st}}$ layer indexed by $S$.

The importance function $\hat{\mathcal{L}}$ is not only designed based on the interdependence between the parameters of the $l^{th}$ and $(l+1)^{st}$ convolutional layers, but it is also formulated to ensure minimal impact on network performance. To demonstrate this minimal impact, the following theorem provides an upper bound on the average absolute error in the output of $(l+1)^{st}$ convolutional layer when the $p_r^{th}$ filter is pruned from the $l^{th}$ layer. The proof is provided in supplementary material.

**Theorem 4.1.** *In a CNN, let $X_{l+1}$ be the output of the $(l+1)^{st}$ convolutional layer and $\hat{X}_{l+1}^{p_r}$ be the output when $p_r^{th}$ filter from the $l^{th}$ layer ($l = 1, \ldots, \mathbb{L}$) is pruned. The average absolute error in the output of the $(l+1)^{st}$ convolutional layer is bounded above as follows*

$$AAE(X_{l+1}) = E(\| X_{l+1} - \hat{X}_{l+1}^{p_r} \|_2^2)$$
$$\leq \| F^{l+1}[:,:,p_r,:] \|_2^2 \| F^l[:,:,:,p_r] \|_2^2 \, E(\Lambda_l). \tag{9}$$

where $\Lambda_l$ depends solely on the parameters of the convolutional layers preceding the $l^{th}$ layer and on the input dataset. For more details on $\Lambda_l$, see appendix 7.6.

### 4.1 $\gamma$-WEAKLY SUBMODULARITY OF THE PROPOSED METHOD

In the following theorem, we prove the submodularity of our importance function. This characteristic allows us to employ an oblivious algorithm for filter selection while achieving a reliable approximation. The detailed definition of submodularity is given in appendix 7.3.

**Theorem 4.2.** *The importance function $\hat{\mathcal{L}}$ defined in Equation equation 8 is a $\gamma_{U,k}$-weakly submodular function.*

*Proof:* see appendix 7.4. □

*Note*: In this paper, we assume that for each convolutional layer $l$ ($l = 1, \ldots, \mathbb{L}$) and all indices $i \in \{1, 2, \ldots, H_l\}$, at least one of the norms $\|F^l_{\{i\}F}\|_2^2$ or $\|F^{l+1}_{\{i\}C}\|_2^2$ is nonzero. If both are zero, the filter $F^l_{\{i\}F}$ and its corresponding channels in the $(l+1)^{st}$ layer, $F^{l+1}_{\{i\}C}$, can be eliminated as their presence has no impact on the network.

As demonstrated, our proposed importance function $\hat{\mathcal{L}}$ constitutes a $\gamma$-weakly submodular function, as proven in Theorem 4.2. In the following section, we advocate utilizing the oblivious algorithm to address this weakly submodular maximization for filter selection. Notably, the convergence of this algorithm for weakly submodular functions is assured (Das and Kempe, 2011).

---

**Algorithm 1** Oblivious Algorithm for Selecting Filters to Prune from the $l^{\text{th}}$ Convolutional Layer

---

**Input:** Filters in the $l^{\text{th}}$ and $(l+1)^{\text{st}}$ convolutional layers, pruning rate $r$
**Output:** $S^{\text{obv}}$, the set of selected filters in the $l^{\text{th}}$ layer for removal
1: $I \leftarrow \{1, 2, \ldots, H_l\}$
2: $S^{\text{obv}} \leftarrow \emptyset$
3: **for all** $p \in \{1, \ldots, H_l\}$ **do**
4: $\quad \mathcal{L}_{\text{pr}}(p) \leftarrow \max \left( \left\| F^l_{\{p\}F} \right\|^2_2, \left\| F^{l+1}_{\{p\}C} \right\|^2_2 \right)$
5: **end for**
6: **while** $|S^{\text{obv}}| \leq r \cdot H_l$ **do**
7: $\quad$ Compute index $p^*$ using Equation equation 10
8: $\quad S^{\text{obv}} \leftarrow S^{\text{obv}} \cup \{p^*\}$
9: $\quad I \leftarrow I \setminus \{p^*\}$
10: **end while**

---

## 4.2 Oblivious Pruning Algorithm

To major type of algorithm used in filter pruning are greedy and oblivious. In this section, we employ the oblivious algorithm to tackle optimization problem equation 7 and benefit from its simplicity and low-complexity precess. Algorithm 1 presents the oblivious algorithm for selecting the set $S^{obv}$ of filters for pruning from $l^{th}$ convolutional layer, utilizing the proposed method.

The steps for computing the importance of a single filter in $l^{th}$ convolutional layer is outlined as follows.

- The $l_2$-norm of each filter $F^l_{\{p\}F}$ and the tensor of its corresponding channels $F^{l+1}_{\{p\}C}$ is computed, for $p = 1, \ldots, H_l$.

- The value $\hat{\mathcal{L}}(p)$ is determined, i.e., the maximum of $\|F^l_{\{p\}F}\|^2_2$ and $\|F^{l+1}_{\{p\}C}\|^2_2$ for $p = 1, \ldots, H_l$.

- The index $p^*$ is chosen for which $\hat{\mathcal{L}}(p^*)$ is minimum in the set $\{\hat{\mathcal{L}}(p)|p = 1, \ldots, H_l\}$.

- The filter $F^l_{\{p^*\}F}$ in the $l^{th}$ convolutional layer is selected for removal.

To generalize this procedure for selecting $i^{th}$ filter $F^l_{\{p^*_i\}F}$ for removal, we use

$$p^*_i = \arg\min_{p \in I} \hat{\mathcal{L}}(p), \tag{10}$$

where $i = 1, \ldots, k$, and $I = \{1, \ldots, H_l\} \setminus \{p^*_1, \ldots, p^*_{i-1}\}$. In the following theorem, we establish that Algorithm 1 provides an approximation guarantee denoted by $\gamma_{\emptyset,k}$.

We establish in Theorem 4.3 that the $\gamma_{U,k}$-weak submodularity of the importance function $\hat{\mathcal{L}}$, as defined in equation 8, guarantees the solution obtained by this algorithm, closely approximating the optimal value with an error bounded by $\frac{\gamma_{\emptyset,k}}{k}$, where $\emptyset$ represents the empty set.

**Theorem 4.3.** *The set $S^{obv}$ of size $k$ selected by the oblivious algorithm has the following approximation guarantees:*

$$\hat{\mathcal{L}}(S^{obv}) \geq \frac{\gamma_{\emptyset,k}}{k} \hat{\mathcal{L}}(S^*), \tag{11}$$

*and*

$$\hat{\mathcal{L}}(S^{obv}) + \mathcal{R}_{obv} \geq \gamma_{\emptyset,k} \hat{\mathcal{L}}(S^*), \tag{12}$$

*where $S^*$ is the optimal solution of the problem equation 7, and $\mathcal{R}_{obv}$ is defined as $\| F^l_{S^{obv}_1 F} \|^2_2 - \| F^{l+1}_{S^{obv}_1 C} \|^2_2$ or $\| F^{l+1}_{S^{obv}_2 C} \|^2_2 - \| F^l_{S^{obv}_2 F} \|^2_2$, depending on whether $\hat{\mathcal{L}}(S^{obv})$ is given by $\| F^l_{S^{obv} F} \|^2_2$ or $\| F^{l+1}_{S^{obv} C} \|^2_2$, respectively, and the sets $S^{obv}_1$ and $S^{obv}_2$ are defined as follows:*

$$S^{obv}_1 = \{i \in S \mid \| F^l_{\{i\}F} \|^2_2 \geq \| F^{l+1}_{\{i\}C} \|^2_2\}, \tag{13}$$

*and*

$$S^{obv}_2 = \{i \in S \mid \| F^l_{\{i\}F} \|^2_2 \leq \| F^{l+1}_{\{i\}C} \|^2_2\}. \tag{14}$$

*Proof:* see Appendix 7.5. □

The Equation equation 11 illustrates that the set of indices selected by the oblivious algorithm is close to the optimal set $S^*$ by an approximation of $\frac{\gamma_{\emptyset,k}}{k}$. Moreover, Equation equation 12 gives this approximation by two parameters $\gamma_{\emptyset,k}$ and $\mathcal{R}_{obv}$. The closer $\gamma_{\emptyset,k}$ to 1 and $\mathcal{R}_{obv}$ to zero the closer the selected set to the optimal solution set. As seen, computing $\gamma_{\emptyset,k}$ involves evaluating an NP-hard combinatorial problem over all filter subsets of size at most $k$, totaling $\sum_{i=0}^{k} \binom{H_l}{i}$ subsets. For instance, the first layer of VGG-16 (with 64 filters) has over 8 million such subsets when $k = 5$. For example, if in the layer $l$ of a CNN, the norm of every filter is equal to the norm of its corresponding channels in the layer $l + 1$, then $\gamma_{\emptyset,k} = 1$, and $\mathcal{R}_{obv} = 0$, therefore the solution of the oblivious algorithm is exactly the optimal solution set, see Appendix 7.4 for more examples.

## 5    EXPERIMENTS

In this section, we evaluate the performance of pruning methods under data-free conditions. Specifically, we investigate the efficacy of one-shot compression, where a pre-trained model undergoes complete compression in a single step. The proposed pruning method is followed by fine-tuning to restore the network's accuracy.

### 5.1    RCE EVALUATION

To evaluate the carbon footprint of our method and compare it with baseline methods, we consider the hardware and software specifications of our system (detailed in Appendix 7.9). The system is located in Québec, Canada, where the carbon intensity is approximately 1.2 grams of $CO_2$e per kilowatt-hour.

For example, pruning 50% of the filters in ResNet-56 using our method requires 1,659,312 FLOPs. Given the system's compute capability of $13.4 \times 10^{12}$ FLOP/s, the estimated runtime is:

$$\text{Runtime} = \frac{\text{FLOPs}}{\text{FLOP/s}} = 1.2 \times 10^{-7} \text{ seconds.}$$

This corresponds to an estimated carbon footprint of approximately $2.5 \times 10^{-9}$ grams of $CO_2$, which is negligible. In comparison, the estimated footprint of APIB (Guo et al., 2023) under identical system and location conditions is 2.5 grams of $CO_2$ for the same pruning task—a difference of several orders of magnitude. Following Equation equation 3 and the runtime formulation above, the relative carbon efficiency between two methods can be approximated by the ratio of their FLOP requirements. For our method and APIB, this ratio is approximately $1.1 \times 10^{-9}$.

Table 1 reports the pairwise ratios of the total number of operations required by various pruning baselines, including our method with both greedy and oblivious algorithms. Each entry in the $i^{\text{th}}$ row and $j^{\text{th}}$ column is given by $\frac{N_R(i)}{N_C(j)}$, where $N_R(i)$ and $N_C(j)$ denote the total number of operations required by the methods in the respective row and column. As shown in the first row, the estimated carbon footprint of our oblivious algorithm is only $3.62 \times 10^{-3}$ times that of the greedy variant. The reduction is even more pronounced when compared to other baselines: $4.25 \times 10^{-8}$ (F-ThiNet), $2.04 \times 10^{-9}$ (ThiNet), $2.01 \times 10^{-6}$ (HRank), and $9.09 \times 10^{-10}$ (APIB). This demonstrates that filter selection process of the proposed method is millions to billions of times more environmentally efficient.

Conversely, the sixth row of the table shows that APIB incurs a carbon footprint nearly $10^9$ times larger than our method, underscoring the dramatic disparity in environmental cost. Other methods are also significantly less efficient: Greedy ($2.76 \times 10^2$), F-ThiNet ($2.35 \times 10^7$), ThiNet ($4.91 \times 10^8$), and HRank ($4.97 \times 10^5$). These results highlight the substantial carbon reduction potential enabled by our low-complexity pruning framework. The detailed procedure for computing the number of operations across baselines is provided in Appendix 7.6.1.

### 5.2    RESOURCE EFFICIENCY

Tables 2 presents the RE for the proposed method and the main baselines, HRank (Lin et al., 2020), ThiNet (Luo et al., 2018), IPBM (Guo et al., 2023), and RED++ (Yvinec et al., 2023), applied to

Table 1: RCE of the proposed method compared to other pruning techniques.

| | Ours (Oblivious) | Ours (Greedy) | F-ThiNet | ThiNet | HRank | APIB | RED++ |
|---|---|---|---|---|---|---|---|
| **Ours (Obl)** | 1 | 3.62e-03 | 4.25e-08 | 2.04e-09 | 2.01e-06 | 9.94e-10 | 7.28e-02 |
| **Ours (Gr)** | 2.76e+02 | 1 | 1.17e-05 | 5.62e-07 | 5.56e-04 | 2.75e-07 | 2.01e+01 |
| **F-ThiNet** | 2.35e+07 | 8.53e+04 | 1 | 4.79e-02 | 4.74e+01 | 2.34e-02 | 1.71e+06 |
| **ThiNet** | 4.91e+08 | 1.78e+06 | 2.09e+01 | 1 | 9.88e+02 | 4.88e-01 | 3.58e+07 |
| **HRank** | 4.97e+05 | 1.80e+03 | 2.11e-02 | 1.01e-03 | 1 | 4.94e-04 | 3.62e+04 |
| **APIB** | 1.01e+09 | 3.64e+06 | 4.27e+01 | 2.05e+00 | 2.02e+03 | 1 | 7.32e+07 |
| **RED++** | 1.37e+01 | 4.97e-02 | 5.83e-07 | 2.80e-08 | 2.76e-05 | 1.37e-08 | 1 |

ResNet-50 on ImageNet. In this experiment, the accuracy of the pruning methods is sourced from the literatures. Since the pruning ratios corresponding to each accuracy are not explicitly provided in the literature, we approximate them using the number of parameters reduced for each accuracy. RED++ offers graphical results for these networks, therefore, we estimate the accuracy preservation for RED++ to be around 95.5%. The percentage of parameters removed is also approximated from the graphs in literature. When calculating the MAC for RED++, we only account for the computations involved in the redundancy evaluation step during the similarity-based clustering. As a result, the actual MAC of RED++ is likely higher than the values presented here.

Table 2: Resource Efficiency (RE) of pruning methods on *ResNet-50 (ImageNet)* and *ResNet-56 (CIFAR-10)*.

| Method | ResNet-50 (ImageNet) | | ResNet-56 (CIFAR-10) | |
|---|---|---|---|---|
| | **Pruning Ratio (r)** | **RE** | **Pruning Ratio (r)** | **RE** |
| Ours | 0.20 | 4.4e-07 | 0.09 | 5.7e-05 |
| | 0.32 | 4.3e-07 | 0.17 | 5.7e-05 |
| | 0.72 | 4.0e-07 | 0.25 | 5.6e-05 |
| | | | 0.50 | 5.5e-05 |
| | | | 0.60 | 5.5e-05 |
| RED++ (Yvinec et al., 2023) | 0.33 | 2.1e-09 | 0.68 | 4.1e-06 |
| HRank (Lin et al., 2020) | 0.25 | 1.0e-12 | 0.09 | 1.1e-10 |
| | 0.35 | 9.9e-13 | 0.25 | 1.1e-10 |
| | 0.60 | 9.5e-13 | 0.48 | 1.2e-11 |
| IPBM (Guo et al., 2023) | 0.39 | 1.1e-16 | 0.33 | 5.6e-14 |
| | 0.48 | 1.1e-16 | 0.43 | 5.6e-14 |
| | | | 0.57 | 5.5e-14 |
| ThiNet | 0.30 | 6.6e-18 | | |
| | 0.50 | 4.3e-18 | | |
| | 0.70 | 3.4e-18 | | |

The results presented in Tables 2 demonstrate that the resource efficiency of each pruning method remains relatively consistent across different pruning ratios. For instance, the RE of the proposed method on ResNet-50 is approximately 4e-7, regardless of the pruning ratio, within the range of 0.2 to 0.72. This trend is similarly observed across other networks and pruning methods suggesting that RE serves as a reliable metric for evaluating the computational cost of a pruning method relative to the accuracy achieved. Based on the RE values presented in Tables 2 the pruning method closest in performance to our method is RED++, which consistently achieves approximately one-tenth of our method 's resource efficiency across all tested networks. In comparison, the resource efficiency of IBPM is substantially lower, with the proposed method demonstrating an RE approximately e+09 times higher. Similarly, HRank exhibits a much lower RE, with ours outperforming it by a factor of e+05. These results highlight the significant advantage of the proposed method in terms of resource efficiency, underscoring its superior balance between network accuracy preservation and computational complexity.

## 5.3 EXPERIMENTAL RESULTS

We employ commonly used protocols, namely the number of parameters and required Floating Point Operations (FLOPs), to assess model size and computational demands. Our experiments are carried out on two most popular datasets CIFAR-10 (Krizhevsky et al., 2009) and ILSVRC-2012 (Deng et al., 2009). To assess the efficacy and performance of our proposed pruning method, three of the most used networks ResNet-50, ResNet-56, and ResNet-110 are considered, utilizing ResNet-50 on ILSVRC-2012 and ResNet-56 and ResNet-110 on CIFAR-10. In order to evaluate the performance of the pruned networks, the Top-1 and/or Top-5 accuracies are provided.

Table 3: Compression results of ResNet-50 on ILSVRC-2012.

| Model | Parameters | FLOPs | Top1 (%) | Top-5 (%) |
|---|---|---|---|---|
| Baseline (He et al., 2016) | 25.5M | 4.09B | 76.15 | 92.87 |
| DECORE (Lin et al., 2019) | 22.69M | 3.54B | 76.31 | 93.02 |
| GAL-0.5 (Lin et al., 2019) | 21.20M | 2.33B | 71.95 | 90.94 |
| PFP (Liebenwein et al., 2020) | 20.88M | 3.65B | 75.91 | 92.81 |
| GAL-0.5 joint (Lin et al., 2019) | 19.31M | **1.84B** | 71.80 | 90.82 |
| SSS-32 (Huang and Wang, 2018) | 18.60M | 2.82B | 74.18 | 91.91 |
| **Ours** | **18.22M** | 2.86B | **76.52** | **93.18** |
| PFP (Liebenwein et al., 2020) | 17.82M | 2.29B | 75.21 | 92.43 |
| HRank (Lin et al., 2020) | 16.15M | 2.30B | 74.98 | 92.33 |
| He et al. (He et al., 2017) | - | 2.73B | 72.30 | 90.80 |
| SSS-26 (Huang and Wang, 2018) | 15.60M | 2.33B | 71.82 | 90.70 |
| GDP-0.6 (Lin et al., 2018) | - | 1.88B | 71.19 | 90.71 |
| GDP-0.5 (Lin et al., 2018) | - | **1.57B** | 69.58 | 90.14 |
| GAL-1 (Lin et al., 2019) | 14.67M | 1.58B | 69.88 | 89.75 |
| FilterSketch (Lin et al., 2021) | 14.53M | 2.23B | 74.68 | 92.17 |
| **Ours** | **14.61M** | 2.26B | **75.37** | **92.56** |
| GAL-1 joint (Lin et al., 2019) | 10.21M | 1.11B | 69.31 | 89.12 |
| ThiNet-50 (Luo et al., 2018) | 8.66M | 1.10B | 68.42 | 88.30 |
| HRank (Lin et al., 2020) | 8.27M | 0.98B | 69.10 | **89.58** |
| FilterSketch (Lin et al., 2021) | 7.18M | 0.93B | 69.43 | 89.23 |
| **Ours** | **6.67M** | **0.96B** | **69.58** | 89.31 |

We compare our proposed pruning method with the baselines including but not limited to HRank (Lin et al., 2020), RED++ (Yvinec et al., 2023), ThiNet (Luo et al., 2018), GAL (Lin et al., 2019), SSS (Huang and Wang, 2018), L1 (;and Hans Peter Graf, 2017), F-ThiNet (Tofigh et al., 2022), He et al., (He et al., 2017), GDP (Lin et al., 2018), NISP (Yu et al., 2018), DECORE (Alwani et al., 2021), FilterSketch (Lin et al., 2021), and APIB (Guo et al., 2023). We present the results of other methods based on their respective literature, marking them as '-' if not reported. The best pruning results are highlighted in bold. In this section, we present the results exclusively for ResNet-50 on ImageNet. The outcomes for other networks and datasets are provided in appendix 7.8.1.

**ResNet-50 on ImageNet:** As shown in Table 3, our approach achieves an outstanding Top-1 accuracy of 76.52%, surpassing the baseline model (76.52% vs. 76.15%) with similar FLOPs and parameter reduction as SSS-32. Notably, our method yields a significantly greater reduction in parameter count and nearly identical reduction in FLOPs compared to FilterSketch, while achieving superior accuracy (75.37% vs. 74.68%). Moreover, even with extensive pruning, our method remains better than top-1 accuracy including GAL-1 joint, ThiNet-50, HRank, and FilterSketch while achieving greater reductions in parameter count and FLOPs (69.58% vs. 69.31% for GAL-1 joint, 68.42% for ThiNet-50, 69.1% for HRank and 69.43% for FilterSketch). Since no numerical results is reported by RED++ (Yvinec et al., 2023) for ResNet-50 on ImageNet, we compare our method with RED++ using their metric: the percentage of preserved accuracy relative to the percentage reduction in parameters. As shown in Table 3, the proposed method preserves almost 100% of the accuracy while achieving a 57% reduction in parameters, outperforming all the results presented in (Yvinec et al., 2023). Moreover, for 28.5% reduction in the parameters, the percentage of the preserved accuracy by ours is around 100.5%.

## 5.4 COMPLEXITY EVALUATION

We compare the complexity of various pruning methods for filter selection in convolutional layers of ResNet-56 on CIFAR-10, presenting the results in Figure 1. The number of operations in the process of the filter selection is used as the complexity metric. In all the results presented in this section, a 50% pruning ratio and a batch size of 256 is considered. For complexity of ThiNet and F-ThiNet methods, the parameter $\mathbb{I}$, the number of entries in the output of $(l+1)^{st}$ layer, is 10.

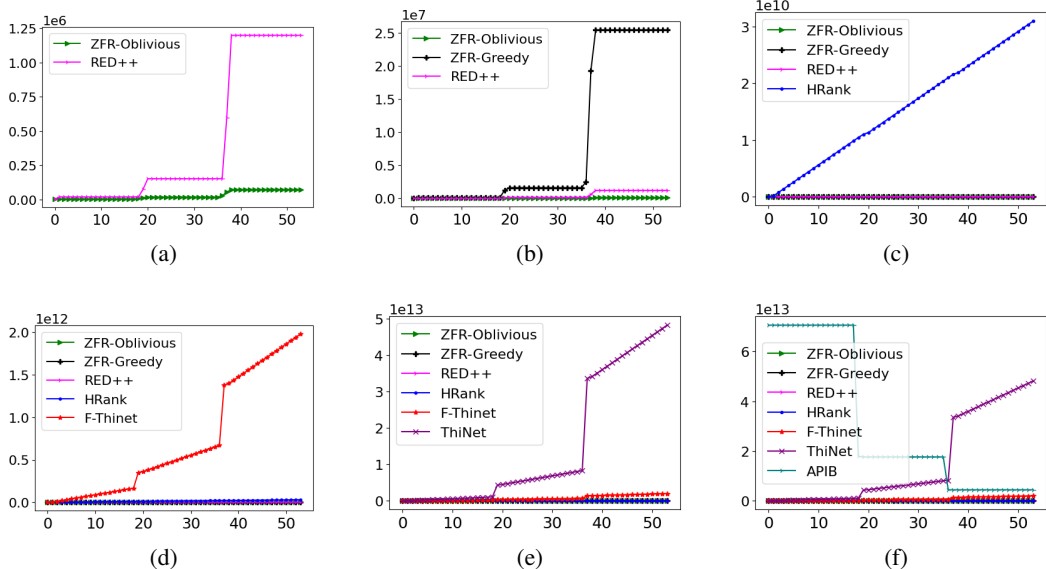

Figure 1: Complexity comparison of different pruning methods for ResNet-56 on CIFAR-10. The X-axis represents the layer number and the Y-axis represents the number of operations required in the filter selection.

Overall, Figure 1 highlights the substantial gap in complexity between our oblivious algorithm and existing pruning methods. As shown in Figure 1a, our method requires only $1.7 \times 10^6$ operations, compared to $2.3 \times 10^7$ for RED++, making RED++ at least 13.7 times more computationally expensive. Figure 1b further compares our oblivious and greedy variants with RED++; the greedy algorithm demands $2.76 \times 10^2$ more operations than the oblivious version, nearly flattening its curve in comparison. Additional results in Figures 1c, 1d, and 1e confirm the higher complexity of HRank, F-ThiNet, and ThiNet. Finally, Figure 1f shows that APIB is the most computationally demanding method, requiring up to $1.01 \times 10^9$ times more operations than our oblivious approach, except in deeper layers where ThiNet is slightly higher.

## 6 CONCLUSION

In this work, we introduced two complementary metrics—**resource efficiency** and **carbon efficiency**—to provide a more comprehensive and sustainability-aware evaluation framework for pruning methods beyond the traditional metrics. Building on these metrics, we proposed a novel filter pruning approach that addresses the inefficiencies of existing strategies. Our method leverages a low-complexity oblivious algorithm grounded in weak submodular optimization, offering both theoretical guarantees and significant reductions in pruning complexity. Extensive empirical evaluations show that our approach not only recovers the accuracy of pruned networks at a state-of-the-art level but also achieves substantial gains in computational efficiency and environmental sustainability. These results highlight the promise of our framework as a practical and sustainable direction for advancing green pruning in deep neural networks.

*We acknowledge the use of large language models (LLMs) to polish the writing of this paper.

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

# 7 APPENDIX

## 7.1 RELATED WORK

In this appendix, we present a comprehensive literature review of various aspects of pruning methods, focusing on how these characteristics impact their efficiency. Table 4 provides a comparison of the main pruning methods based on these characteristics, emphasizing the key strengths of the proposed method.

### 7.1.1 COMPLEXITY OF PRUNING METHODS

The increasing complexity of pruning methods presents a significant challenge for resource-constrained applications, often necessitating the use of one-shot pruning approaches (Lee et al., 2018). Interestingly, in the case of Dense Training Scenarios (DTS), different pruning methods yield comparable performance. Moreover, in the low-density regime, the simplest approach—magnitude-based pruning—demonstrates superior results (Nowak et al., 2023). In general, pruning methods for CNNs face a trade-off: low-complexity methods tend to lack accuracy, while highly accurate methods are typically computationally intensive. The complexity of a pruning technique is primarily determined by its data dependency and the algorithm employed for filter selection. Prior works (;and Hans Peter Graf, 2017), (He et al., 2019), (Lin et al., 2021) exclusively prune filters in a data-free manner, suggesting less complex algorithms, their performance in terms of network accuracy often lags behind data-based approaches (Lin et al., 2020), (Luo et al., 2018), (Lin et al., 2019), (Tofigh et al., 2022), (Alwani et al., 2021), (Qian et al., 2023), (Huang and Wang, 2018), in which dataset is integrated into filter selection alongside network parameters, resulting in a computationally intensive process. Although the pruning methods proposed by (Yvinec et al., 2021) and (Yvinec et al., 2023) are data-free and less complex than their data-based counterparts, they achieve commendable performance. However, their three-step process introduces a level of complexity higher than what is typically expected from a data-free approach. According to the classification by (He and Xiao, 2024), data-free structured pruning methods are categorized into filter norm and filter correlation groups. The filter norm methods, exemplified by (;and Hans Peter Graf, 2017), are more efficient in terms of complexity, as it determines filter importance based solely on the norm. On the other hand, (He et al., 2019), (Yvinec et al., 2023), (Yvinec et al., 2021) fall under the filter correlation category. Notably, all existing data-free pruning methods calculate filter importance based solely on the filters in the current layer, without accounting for the dependencies between these filters and their corresponding channels in adjacent layers. This limitation makes them less comprehensive compared to data-based methods.

Unlike the aforementioned methods in this category, our proposed method offers a more comprehensive filter selection process by accounting for the interdependence between parameters in the pruned layer and its adjacent layers.

### 7.1.2 MATHEMATICAL PERSPECTIVE FOR PRUNING

Most of the proposed pruning methods in the literature are heuristic and lack theoretical guidance. Some prior works have explored a theoretical perspective for pruning networks. Studies (Sarvani et al., 2022), (Guo et al., 2023), (Wang et al., 2020), (Dai et al., 2018), (Zheng et al., 2021) investigate the removal of filters based on *mutual information* (MI) or a variation of MI known as the *information bottleneck* (IB) (Naftali et al., 2000), (Tishby and Zaslavsky, 2015). However, the computation of MI and IB involves density estimation in high-dimensional space, which is computationally infeasible in most practical cases. To cope with this challenge, some of these pruning methods propose new ideas, e.g., (Guo et al., 2023) implements its IB pruning problem by solving a *Hilbert-Schmidt independence criterion lasso* problem under certain conditions, but they still have a high complexity filter selection (Guo et al., 2023). Moreover, (Wang et al., 2020), (Dai et al., 2018), (Sarvani et al., 2022) within this category might struggle to recover their performance after pruning, so, they use special fine-tuning methods to solve the problem. Similarly, the mathematical framework proposed in (Zheng et al., 2021) is not sufficiently helpful to avoid heuristic pruning criteria like $l_1$-norm. Notably, (El Halabi et al., 2022) presents a structured pruning method focused on the limited-data regime that relies on reweighting methods (Mariet and Sra, 2016) and considers submodular optimization. Furthermore, (Yvinec et al., 2023) introduces an upper bound on weight differences to quantify parameter similarity, which enables efficient weight clustering for filter pruning.

In this work, we propose an importance function based on an upper bound of the AAE in the output of the adjacent layer. Additionally, the $\gamma$-weak characteristic of the proposed importance function guarantees the convergence of the oblivious algorithm used for filter selection.

### 7.1.3 GREEDY AND OBLIVIOUS ALGORITHM IN PRUNING METHODS

Greedy algorithms are frequently employed by pruning methods as the means of converging to an optimized solution (Luo et al., 2018), (Guo et al., 2023).While they offer a promising approach for

tackling NP-hard problems, the complexity of greedy algorithms remains high due to their iterative nature, making them less suitable for devices with limited computing resources. To address this, some studies have explored alternative oblivious algorithms, albeit without providing a rigorous mathematical justification (Tofigh et al., 2022), (Lin et al., 2020), (;and Hans Peter Graf, 2017). Although greedy algorithms are grounded in a stronger mathematical foundation, it has been heuristically demonstrated that oblivious algorithms can achieve comparable performance. For instance, in our previous work, F-ThiNet (Tofigh et al., 2022) employs the same filter selection strategy as ThiNet (Luo et al., 2018) but utilizes an oblivious algorithm instead of a greedy one, achieving similar performance with significantly lower complexity. In this work, we employ an oblivious algorithm to develop a low-complexity pruning method tailored for resource-limited devices.

### 7.1.4 IMPORTANCE FUNCTION IN PRUNING METHODS

Structured pruning methods define their importance functions from various perspectives, focusing on different components of convolutional neural networks (He and Xiao, 2024). Weight-dependent importance functions determine filter selection solely based on the filters in convolutional layers. For example, (;and Hans Peter Graf, 2017) employs filter norms as the basis for importance, while FPGM (He et al., 2019) and RED++ (Yvinec et al., 2023) use the median and a three-step pruning, respectively, for computing importance. On the other hand, several methods incorporate dataset information into the importance calculation. Both ThiNet (Luo et al., 2018) and F-ThiNet (Tofigh et al., 2022) select filters for pruning by approximating the absolute error in the output of the adjacent layer, whereas HRank (Lin et al., 2020) relies on the rank of the output as a selection metric. Similarly, CHIP (Sui et al., 2021) and NIPS (Yu et al., 2018) assess filter importance based on activation values obtained during forward propagation. Additionally, some pruning techniques employ sparsity regularizers to induce structured sparsity in networks. Methods such as NS (Liu et al., 2017), GBN (You et al., 2019), PR (Zhuang et al., 2020), and RSNLI (Ye et al., 2018) reduce network size by applying regularizers to batch normalization (BN) parameters. Approaches like SSS (Huang and Wang, 2018) and GAL (Lin et al., 2019) prune by introducing regularizers on additional parameters defined in their filter selection process. Alternatively, optimization tools can be used to determine filter importance. For instance, (Peng et al., 2019), (Wang et al., 2019), and (Nonnenmacher et al., 2022) apply Taylor expansion to approximate the loss function when filters are set to zero, thereby computing their importance.

In this work, we propose a weight-dependent importance function which is based on the norm of the filters in the pruned and adjacent layers. Our importance function, inspired by ThiNet (Luo et al., 2018) and RED++ (Yvinec et al., 2023), controls AAE by minimizing an upper bound on it in the output of the adjacent layer.

Table 4: Summary of the existing pruning methods and ours.

| | Data-Free | Filter Norm | Current Layer | Adjacent Layer | Algorithm |
|---|---|---|---|---|---|
| L1 (;and Hans Peter Graf, 2017) | √ | √ | √ | | Oblivious |
| OURS | √ | √ | √ | √ | Oblivious |
| RED++ (Yvinec et al., 2023) | √ | | √ | | - |
| FPGM (He et al., 2019) | √ | | √ | | - |
| HRank (Lin et al., 2020) | | | | √ | - |
| ThiNet (Luo et al., 2018) | | | | √ | Greedy |
| F-ThiNet (Tofigh et al., 2022) | | | | √ | Oblivious |
| IPBM (Guo et al., 2023) | | | √ | | Greedy |

### 7.2 DEFINITION OF GREEDY ALGORITHM

The set $S$ of indices from $V = \{1, 2, \ldots, d\}$ is selected using a greedy algorithm, as described in the following steps:

1. *Initialization:* Begin with an empty set of selected indices, $S = \emptyset$.

2. *First index selection:*
   - Evaluate the function $\mathcal{F}$ for each index in $V$, i.e., $\mathcal{F}(1), \ldots, \mathcal{F}(d)$.
   - Select index $j^* = \text{argmax}\{\mathcal{F}(1), \ldots, \mathcal{F}(d)\}$ and add it to the set $S$, i.e., $S = \{j^*\}$.

3. *Subsequent index selection:*

---

**Algorithm 2** Greedy Algorithm

---

**Input:** Ground set $V = \{1, 2, \ldots, d\}$, set function $\mathcal{F} : 2^V \to \mathbb{R}_+$, and integer $k \in \mathbb{N}_+$
1: $S \leftarrow \varnothing$
2: **while** $|S| < k$ **do**
3: $\quad j^* \leftarrow \arg \max\limits_{j \in V \setminus S} \mathcal{F}(S \cup \{j\})$
4: $\quad S \leftarrow S \cup \{j^*\}$
5: **end while**
**Output:** $S$

---

- For each $r^{th}$ index, compute the function values $\mathcal{F}(\{i \cup S\})$ for all $i \in V \setminus S$.
- Select the index $j^* = \text{argmax}_{i \in V \setminus S} \mathcal{F}(\{i \cup S\})$ and add it to the set $S$.

4. *Repeat:* Repeat the step 3 until the set $S$ contains $k$ indices, i.e., $|S| = k$.

In each iteration of a greedy algorithm, the most relevant index to the previously selected indices is chosen. Although such a greedy algorithm effectively solves weakly submodular optimization problems and is commonly used by pruning methods (El Halabi et al., 2022), (Luo et al., 2018), its high-complexity iterative procedure has high a high carbon footprint (comparing to oblivious algorithm which selects filters in one iteration) and is challenging for tiny devices.

### 7.3 DEFINITION OF $\gamma-$WEAKLY SUBMODULAR SET FUNCTION

Consider a ground set $V = \{1, 2, \ldots, d\}$ and a set function $\mathcal{F} : 2^V \to \mathbb{R}_+$. We define the marginal gain of including a set $J \subseteq V$ with another set $R \subseteq V$ as $F(J \mid R) = \mathcal{F}(R \cup J) - \mathcal{F}(R)$. This measures the alteration in value upon adding $J$ to $R$. The size of a set $R$ is denoted as $|R|$.

**Definition 7.1.** (El Halabi et al., 2022) Given a set function $\mathcal{F} : 2^V \to \mathbb{R}_+$, a set $U \subseteq V$, and an integer $k \in \mathbb{N}$, we consider that $\mathcal{F}$ is $\gamma_{U,k}$-weakly submodular, with $\gamma_{U,k} > 0$ if

$$\gamma_{U,k} \mathcal{F}(R \mid L) \leq \sum_{i \in R} \mathcal{F}(i \mid L), \tag{15}$$

for every two disjoint sets $L, R \subseteq V$, such that $L \subseteq U$ and $|R| \leq k$. The parameter $\gamma_{U,k}$ is called the submodularity ratio of $\mathcal{F}$.

An efficient method to solve weakly submodular optimization problems and achieve a good approximation to the optimal solution is using a greedy algorithm (Das and Kempe, 2011).

### 7.4 PROOF OF THEOREM 4.2

Given set $V = \{1, 2, \ldots, H_l\}$, the set of indices of the filters in $l^{th}$ convolutional layer, for every $U \subset V$, $L \subset U$, and $i \notin L$

$$\mathcal{L}_{pr}(i|L) = \max\{\| F_{(L \cup \{i\})F}^l \|_2^2, \| F_{(L \cup \{i\})C}^{l+1} \|_2^2\} \tag{16}$$

$$- \max\{\| F_{LF}^l \|_2^2, \| F_{LC}^{l+1} \|_2^2\}$$

$$= \max\{\| F_{LF}^l \|_2^2 + \| F_{\{i\}F}^l \|_2^2 \tag{17}$$

$$, \| F_{LC}^{l+1} \|_2^2 + \| F_{\{i\}C}^{l+1} \|_2^2\} \tag{18}$$

$$- \max\{\| F_{LF}^l \|_2^2, \| F_{LC}^{l+1} \|_2^2\} > 0. \tag{19}$$

Now, let's assume that $R \subset V$ is a set that is disjoint from the set $U$ such that $|R| \leq k$. Therefore, $\sum_{i \in R} \mathcal{L}_{pr}(i|L) > 0$. Setting $\gamma_{U,k}$ as follows

$$\gamma_{U,k} = \min_{L \subset U} \min_{R \subset V, |R| \leq k} \frac{\sum_{i \in R} \mathcal{L}_{pr}(i|L)}{\mathcal{L}_{pr}(R|L)}, \tag{20}$$

the inequality condition in Equation equation 15 is satisfied. Because $U$ has finite subsets, and for every set $L \subset U$ there is a finite number of such sets $R$, so, the feasible set in Equation equation 20

is finite. As a result, a positive number $\gamma_{U,k}$ exists. Since for every set $U \subset \{1, 2, \ldots, H_l\}$ and every integer number $k \in \mathbb{N}^+$, the submodularity ratio $\gamma_{U,k}$ exists, the set function $\mathcal{L}_{pr}$ is a $\gamma_{U,k}$-weakly submodular.

As follows, we explore the submodularity of $\| \cdot \|_2^2$ as a set function on the set of filters in a convolutional layer. For the set of filters in the $l^{th}$ layer of a CNN, the function $\| \cdot \|_2^2$ can be regarded as a set function. Given a subset $S$ of $\{1, 2, \ldots, H_l\}$, the index set of the filters in the $l^{th}$ layer, it returns $\| F^l[:, :, :, S] \|_2^2$. The following lemma can be easily proven for this set function.

**Lemma 7.2.** *For any subset $S \subset \{1, 2, \ldots, H_l\}$ and any index $\alpha \notin S$ we have:*

$$\| F^l[:, :, :, S \cup \{\alpha\}] \|_2^2 - \| F^l[:, :, :, S] \|_2^2 = \| F^l[:, :, :, \alpha] \|_2^2, \tag{21}$$

*and*

$$\| F^{l+1}[:, :, S \cup \{\alpha\}, :] \|_2^2 - \| F^{l+1}[:, :, S, :] \|_2^2 = \tag{22}$$
$$\| F^{l+1}[:, :, \alpha, :] \|_2^2 .$$

The following theorem shows that $\| \cdot \|_2^2$ is a submodular set function.

**Theorem 7.3.** *The submodularity ratio of the set function $\| \cdot \|_2^2$ is 1. The set of size $K$ selected by the oblivious algorithm equipped with the $\| \cdot \|_2^2$, as the importance function, is the optimal solution.*

*Proof:* Let's assume $U \subset V$. For any $L \subset U$ and $S \subset V$ where $|S| \leq k$

$$\gamma_{U,k} = \min_{L \subset U \; S \subset V, |S| \leq k} \left( \frac{\sum_{i \in S} \| i | L \|_2^2}{\| S | L \|_2^2} \right) \tag{23}$$

$$= \min_{L \subset U \; S \subset V, |S| \leq k} \left( \frac{\sum_{i \in S} \| F^l_{(L \cup \{i\})F} \|_2^2 - \| F^l_{LF} \|_2^2}{\| F^l_{(L \cup \{S\})F} \|^2 - \| F^l_{LF} \|^2} \right) \tag{24}$$

$$= \min_{L \subset U \; S \subset V, |S| \leq k} \left( \frac{\sum_{i \in S} \| F^l_{\{i\}F} \|_2^2}{\| F^l_{SF} \|_2^2} \right) = 1. \tag{25}$$

This theorem shows that when the filter selection criteria is only the $l_2$-norm of the filters, the $\gamma_{U,k} = 1$. It means that the $\| \cdot \|_2^2$ is a submodular set function . On the other hand, since

$$\sum_{i \in S} \| F^l_{\{i\}F} \|_2^2 = \| F^l_{SF} \|_2^2, \tag{26}$$

a set $S$ of filters has the maximum $l_2$-norm when the filters in $S$ individually have the greatest norms among the filters in the $l^{th}$ layer. Therefore, the oblivious algorithm equipped with $\| \ldots \|_2^2$ provides the optimal solution to the maximization problem (1) in the paper.

*Remark* 7.4. In a convolutional layer, if $\| F^l_{\{i\}F} \|_2^2 \geq \| F^{l+1}_{\{i\}C} \|_2^2$ for all $i = 1, \ldots, H_l$, then the importance function $\mathcal{L}_{pr}$ is submodular, and $S^{obv} = S^*$.

*Remark* 7.5. In a convolutional layer, if $\| F^l_{\{i\}F} \|_2^2 \leq \| F^{l+1}_{\{i\}C} \|_2^2$ for all $i = 1, \ldots, H_l$, then the importance function $\mathcal{L}_{pr}$ is submodular, and $S^{obv} = S^*$.

## 7.5 PROOF OF THEOREM 4.3

*Note*: Let $S^*$ be the optimal index set of $k$ filters for the optimization problem equation 7. For the importance function $\mathcal{L}_{pr}$ defined in equation 8, the following inequalities hold:

$$\gamma_{\emptyset,k} = \min_{R \subset V, |R| \leq k} \frac{\sum_{i \in R} \mathcal{L}_{pr}(i)}{\mathcal{L}_{pr}(R)} \geq 1, \tag{27}$$

$$\gamma_{\emptyset,k} \leq \frac{\sum_{i \in S^*} \mathcal{L}_{pr}(i)}{\mathcal{L}_{pr}(S^*)}. \tag{28}$$

Let $S^{obv}$ denote the set selected by the oblivious algorithm, and let $S^*$ represent the optimal index set of $k$ filters. By definition of the oblivious algorithm ($\sum_{i \in S^{obv}} \mathcal{L}_{pr}(i) \geq \sum_{i \in S^*} \mathcal{L}_{pr}(i)$), using Equations equation **??** and equation 28, the following inequalities are hold

$$\mathcal{L}_{pr}(S^{obv}) \geq \frac{\sum_{i \in S^{obv}} \mathcal{L}_{pr}(i)}{k} \geq \frac{\sum_{i \in S^*} \mathcal{L}_{pr}(i)}{k} \geq \frac{\gamma_{\emptyset,k}}{k} \mathcal{L}_{pr}(S^*). \tag{29}$$

which proves Equation equation 11. To prove Equation equation 12, without loss of generality, we assume $\mathcal{L}_{pr}(S^{obv}) = \| F^l_{S^{obv}F} \|_2^2$, therefore

$$\mathcal{L}_{pr}(S^{obv}) = \| F^l_{S^{obv}F} \|_2^2 \tag{30}$$

$$= \| F^l_{S_1^{obv}F} \|_2^2 + \| F^l_{S_2^{obv}F} \|_2^2 \tag{31}$$

$$\leq \| F^l_{S_1^{obv}F} \|_2^2 + \| F^{l+1}_{S_2^{obv}C} \|_2^2 . \tag{32}$$

Since $\mathcal{L}_{pr}(S^{obv}) = \| F^l_{S^{obv}F} \|_2^2$, consequently

$$\| F^l_{S_1^{obv}F} \|_2^2 + \| F^l_{S_2^{obv}F} \|_2^2 \geq \| F^{l+1}_{S_1^{obv}C} \|_2^2 + \| F^{l+1}_{S_2^{obv}C} \|_2^2$$

$$\Rightarrow 2 \| F^l_{S_1^{obv}F} \|_2^2 + \| F^l_{S_2^{obv}F} \|_2^2 - \| F^{l+1}_{S_1^{obv}C} \|_2^2$$

$$\geq \| F^l_{S_1^{obv}F} \|_2^2 + \| F^{l+1}_{S_2^{obv}C} \|_2^2 \tag{33}$$

$$\Rightarrow \mathcal{L}_{pr}(S^{obv}) + (\| F^l_{S_1^{obv}F} \|_2^2 - \| F^{l+1}_{S_1^{obv}C} \|_2^2)$$

$$\geq \sum_{i \in S^{obv}} \mathcal{L}_{pr}(i) \geq \sum_{i \in S^*} \mathcal{L}_{pr}(i) \geq \gamma_{\emptyset,k} \mathcal{L}_{pr}(S^*), \tag{34}$$

now by defining $\mathcal{R}_{obv} = \| F^l_{S_1^{obv}F} \|_2^2 - \| F^{l+1}_{S_1^{obv}C} \|_2^2$, Equation equation 12 is proven. Similarly by assuming $\mathcal{L}_{pr}(S^{obv}) = \| F^{l+1}_{S^{obv}C} \|_2^2$ the inequality hold for $\mathcal{R}_{obv} = \| F^{l+1}_{S_2^{obv}C} \|_2^2 - \| F^l_{S_2^{obv}F} \|_2^2$.

## 7.6 PROOF OF THEOREM 4.1

In this paper, we use the regular definition of convolution operation in a convolutional layer. The $[m, n, s]$ entry in the output of $l^{th}$ convolutional layer is obtained by convolving the $s^{th}$ filter, $F^l_{sF}$, by part of the input, $\bigoplus_l \overleftarrow{(m, n, s)}$. The convolution operation is the inner product of $F^l_{sF}$ and $\bigoplus_l \overrightarrow{(m, n, s)}$. In order to proof the theorem, we calculate the absolute error in the output of the $(l+1)^{st}$ layer. The $[m, n, s]$ entry in the output of the $(l+1)^{st}$ layer is as follows

$$X_{l+1}[m, n, s] = \left\langle F^{l+1}_{\{s\}F}, ReLu\left(\bigoplus_l \overleftarrow{(m, n, s)}\right) \right\rangle = \sum_{p=1}^{H_l} \sum_{i,j=1}^{k_{l+1}} f^{l+1}_{(i,j,p,s)} ReLu(\mathcal{A}^{m,n,s}_{i,j,p}), \tag{35}$$

where

$$\mathcal{A}^{m,n,s}_{i,j,p} = \left(\bigoplus_l \overleftarrow{(m, n, s)}\right)[i, j, p] = \left\langle F^l_{pF}, Relu\left(\bigoplus_{l-1} \overrightarrow{\left(\bigoplus_l \overleftarrow{(m, n, s)}[i, j, p]\right)}\right) \right\rangle. \tag{36}$$

Similarly, $\hat{X}^{p_r}_{l+1}[m, n, s]$, the $[m, n, s]$ entry of the $X_{l+1}$ when the $p_r^{th}$ filter from the $l^{th}$ layer is pruned, is as follows

$$\hat{X}^{p_r}_{l+1}[m, n, s] = \sum_{p=1}^{p_r-1} \sum_{i,j=1}^{k_{l+1}} f^{l+1}_{(i,j,p,s)} ReLu(\mathcal{A}^{m,n,s}_{i,j,p}) + \sum_{p=1}^{p_r+1} \sum_{i,j=1}^{k_{l+1}} f^{l+1}_{(i,j,p,s)} ReLu(\mathcal{A}^{m,n,s}_{i,j,p}) \tag{37}$$

Then the absolute error in the the $[m, n, s]$ entry of $X_{l+1}$ before and after pruning is calculated as follows

$$|X_{l+1}[m, n, s] - \hat{X}_{l+1}^{p_r}[m, n, s]|^2 = \sum_{i,j=1}^{k_{l+1}} f_{(i,j,p_r,s)}^{l+1} ReLu(\mathcal{A}_{i,j,p}^{m,n,s})$$

$$\leq \left( \sum_{i,j=1}^{k_{l+1}} \left( f_{(i,j,p_r,s)}^{l+1} \right)^2 \right) \left( \sum_{i,j=1}^{k_{l+1}} (\mathcal{A}_{i,j,p}^{m,n,s})^2 \right) \quad (38)$$

$$\leq \| F_{(:,:,p_r,s)}^{l+1} \|_2^2 \| F_{(:,:,:,p_r)}^{l} \|_2^2 \, \alpha_{(m,n)}^{l-1}, \quad (39)$$

where

$$\alpha_{(m,n)}^{l-1} = \sum_{i=1}^{k_{l+1}} \sum_{j=1}^{k_{l+1}} \left( \| Relu \left( \overset{\leftarrow \quad \longrightarrow}{\bigoplus_{l-1} \left( \bigoplus_{l} \overleftarrow{(m, n)}[i, j, p_r] \right)} \right) \|_2^2 \right). \quad (40)$$

Equations equation 38 and equation 39 can be hold from Cauchy-Schwarz inequality and lemma 1 proven in (**?**).

Therefore, the inequality in Theorem 1 holds by summing over the values of $m$, $n$, and $s$ as follows

$$\| X_{l+1} - \hat{X}_{l+1}^{p_r} \|_2^2 = \sum_{s=1}^{H_{l+1}} \sum_{m=1}^{W_{l+1}} \sum_{n=1}^{L_{l+1}} |X_{l+1}[m, n, s] - \hat{X}_{l+1}^{p_r}[m, n, s]|^2 \quad (41)$$

$$\leq \sum_{s=1}^{H_{l+1}} \sum_{m=1}^{W_{l+1}} \sum_{n=1}^{L_{l+1}} \| F_{(:,:,p_r,s)}^{l+1} \|_2^2 \| F_{(:,:,:,p_r)}^{l} \|_2^2 \, \alpha_{(m,n)}^{l-1} \quad (42)$$

$$= \| F_{(:,:,p_r,:)}^{l+1} \|_2^2 \| F_{(:,:,:,p_r)}^{l} \|_2^2 \, \Lambda_l, \quad (43)$$

where

$$\Lambda_l = \sum_{m=1}^{W_{l+1}} \sum_{n=1}^{L_{l+1}} \alpha_{(m,n)}^{l-1}. \quad (44)$$

### 7.6.1 PROCEDURE OF COUNTING THE FLOPs REQUIRED IN FILTER SELECTION OF DIFFERENT BASELINES

In this appendix, we discuss the procedure of counting the number of multiplication in the filter selection for the pruning methods in Table 5.

Table 5: Complexity estimation for selecting $k$ filters in $l^{th}$ convolutional layer.

| Pruning Method | Complexity |
|---|---|
| Ours -Oblivious | $\approx H_l(n_l^1 + m_l^1)$ |
| Ours -Greedy | $\approx \sum_{i=1}^{k} (H_l - i + 1)[n_l^i + m_l^i]$ |
| RED++ (Yvinec et al., 2023) | $\approx \binom{H_l}{2} n_l^1$ |
| HRank (Lin et al., 2020) | $\approx \mathcal{B}\left( \left( \sum_{i=1}^{l-1} C_{i,H_i} \right) + H_l R_l \right)$ |
| F-ThiNet (Tofigh et al., 2022) | $\approx \mathcal{B} H_l N_{l,1,\mathbb{I}}$ |
| ThiNet (Luo et al., 2018) | $\approx \mathcal{B} \sum_{i=1}^{k} (H_l - i + 1) N_{l,i,\mathbb{I}}$ |
| APIB (Guo et al., 2023) | $\approx \left[ \left( \mathcal{B} \sum_{i=1}^{l-1} C_{i,H_i} \right) + 2\mathcal{B}^2 \mathcal{D}_l + 4\mathcal{B}^3 \right]$ |

**Our Proposed Method - Oblivious 1:** In order to calculate the importance of the filters in the $l^{th}$ layer, regardless of the number of selected filters ($K$), the $l_2$-norm of the filters in the $l^{th}$ layer and the $l_2$-norm of their corresponding channels of the filters in the $(l + 1)^{st}$ layer must be computed. In this paper, to reduce the number of multiplications, instead of the $l_2$-norm, we use the square of the $l_2$-norm.

Calculating the square of the $l_2$-norm of a filter of size $k_l \times k_l \times H_{l-1}$ requires $k_l^2 H_{l-1}$ multiplications. Therefore, calculating the square of the $l_2$-norm of the filters in the $l^{th}$ layer and the square

of the $l_2$-norm of the corresponding channels in the filters of the $(l+1)^{\text{st}}$ layer requires $k_l^2 H_{l-1} H_l$ and $k_{l+1}^2 H_l H_{l+1}$ multiplications, respectively. Thus, the number of multiplications required by the oblivious algorithm equipped with our proposed importance function is $H_l(n_l^1 + m_l^1)$.

**Our Proposed Method - Greedy:** In the greedy algorithm equipped with our proposed importance function, the filters are selected in an iterative manner. In the $i^{th}$ iteration, knowing that already $i-1$ filters have been selected in the previous iterations as a set $S_{i-1}$, the next filter is selected by calculating $\mathcal{L}(S_{i-1} \cup j)$ for all $j \in V \setminus S_{i-1}$, using the importance function equation 8. The number of multiplications required to calculate the importance of $S_{i-1} \cup j$ is $n_l^i + m_l^i = k_l k_l H_{l-1} i + k_{l+1} k_{l+1} i H_{l+1}$. Since in the $i^{th}$ iteration there are $(H_l - i + 1)$ such filters, the total number of the multiplications required by the greedy algorithm for selecting $K$ filter for pruning is $\sum_{i=1}^{K} (H_l - i + 1)[n_l^i + m_l^i]$.

**F-ThiNet:** In this method, for an input image to the network, the summation of squares for a set of values in the output of the $(l+1)^{\text{st}}$ layer is calculated when only a portion of filters exist in the $l^{\text{th}}$ layer. If a set of size $\mathcal{B}$ of the images is used in the filter selection process, the computed number of multiplications for one image is multiplied by $\mathcal{B}$. It is important to mention that, to obtain the output in the $(l+1)^{\text{st}}$ layer, the input image must pass through all the previous layers, requiring all the convolution multiplications. Therefore, calculating the number of multiplications, $C_{i,H_i}$, needed to produce the output of the $i^{\text{th}}$ layer for one input image is necessary.

**HRank:** This Method uses the rank of the channels of the output of the $i^{th}$ layer as the importance function. To calculate the rank of the output, the input image passes though all the layers. In order to compute the number of the multiplications required by the HRank for filter selection, similar to F-ThiNet and ThiNet, $C_{i,H_i}$ in needed to obtain the output of the $i^{th}$ layer. In the $l^{th}$ layer, for one input image, the calculation of the rank for all the output channels needs

$$\sum_{j=1}^{\min(W_l, L_l)-1} (W_l - j)(L_l - j),$$

multiplications.

**ThiNet:** In this method, the computation of the number of the multiplications is the same as F-ThiNet but in the iterative process of the greedy algorithm.

**APIB:** In this method, in order to select filters for pruning from the $l^{th}$ layer, we need to calculate matrices $\bar{L}$, $K^{(k)}$, using the Gaussian kernel function and also matrices $\Gamma$, and $\hat{K}^{(k)}$ using matrix multiplication. All these matrices are for a batch of size $\mathcal{B}$. Similar to the other data-based methods, we also need to pass the input images to the network all the way to the $l^{th}$ layer and all the multiplications must be considered. Note that, to simplify our calculation, we do not even consider the cost of vectorization and solving the optimization problem associated with this method for filter selection.

## 7.7 CONTRIBUTION OF $\| F_{\{p\}T}^l \|_2^2$ AND $\| F_{\{p\}C}^{l+1} \|_2^2$ IN IMPORTANCE FUNCTION $\mathcal{L}_{pr}$

In the oblivious Algorithm 1, the importance function $\mathcal{L}_{pr}$ uses the maximum of $\| F_{\{p\}T}^l \|_2^2$ and $\| F_{\{p\}C}^{l+1} \|_2^2$ as the importance value for filter $F_{\{p\}T}^l$. To demonstrate that both norms play a significant role in the filter selection process, we compare the number of filters whose importance values are derived from $\| F_{\{p\}T}^l \|_2^2$ with those derived from $\| F_{\{p\}C}^{l+1} \|_2^2$. Figure 2 illustrates this comparison, showing the number of filters whose importance is determined by the norm of $F_{\{p\}T}^l$ (i.e., $\| F_{\{p\}T}^l \|_2^2 \geq \| F_{\{p\}C}^{l+1} \|_2^2$) versus those determined by the norm of $F_{\{p\}C}^{l+1}$ (i.e., $\| F_{\{p\}C}^{l+1} \|_2^2 \geq \| F_{\{p\}T}^l \|_2^2$) across the convolutional layers in ResNet-50 pre-trained on ImageNet.

For example, in the second layer of ResNet-50, 75% of filters attain their importance value based on the norm of the corresponding channels in the third layer, $F_{\{p\}C}^3$. According to the results presented by Figure 2, both the norms in importance function $\mathcal{L}_{pr}$ contribute effectively in assigning importance value to the filters.

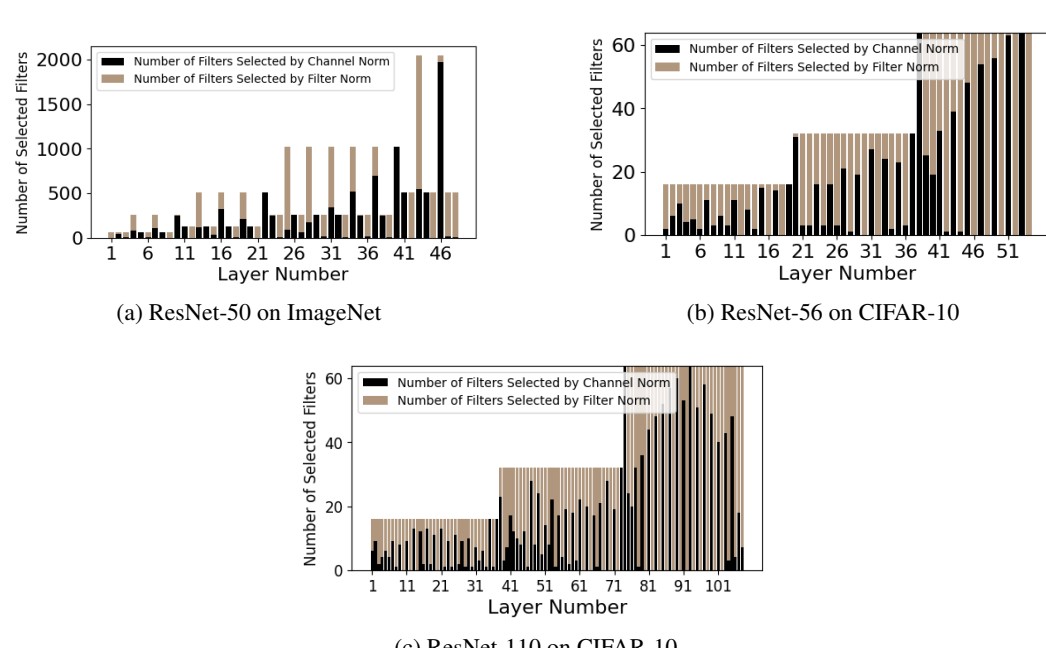

(a) ResNet-50 on ImageNet  (b) ResNet-56 on CIFAR-10

(c) ResNet-110 on CIFAR-10

Figure 2: The layer-wise contribution of the two types of parameters $F^l_{\{i\}T}$ and $F^{l+1}_{\{i\}C}$ in filter selection by $\hat{\mathcal{L}}$ utilizing the oblivious algorithm. The stacked bars compare the number of filters selected by the norm of $F^l_{\{i\}T}$ with the number of filters selected by the norm of $F^{l+1}_{\{i\}C}$.

## 7.8 MORE COMPARISON RESULTS

### 7.8.1 ACCURACY COMPARISON FOR THE PROPOSED METHOD

In this appendix, we provide the results for performance evaluation of our proposed method in ResNet-56 and ResNet-110 networks on CIFAR-10 dataset.

**ResNet-56 on CIFAR-10:** Consistent with our findings on ResNet-50, our method demonstrates an accuracy improvement over the baseline model (94.09% vs. 93.26%) while achieving approximately 16.8% and 17% reduction in FLOPs and parameters, respectively. Furthermore, compared to HRank, FilterSketch, and DECORE, which offer marginally better complexity reduction, our method consistently delivers superior accuracy (93.84% vs. 93.52% for HRank, 93.65% for FilterSketch, and 93.34% for DECORE). Notably, for extensive pruning, our method reduces model complexity compared to APIB (84% vs. 83% in terms of parameters, and 84% vs. 81% in terms of FLOPs) with similar accuracy of the pruned network.

**ResNet-110 on CIFAR-10:** The proposed method consistently outperforms other approaches across all evaluated scenarios, excelling in both performance and complexity reduction. Our method achieves a 59.7% reduction in parameters while also delivering a 0.41% accuracy increase compared to the leading pruning methods, HRank (Lin et al., 2020) and DECORE (Alwani et al., 2021). When compared to other methods like L1 (;and Hans Peter Graf, 2017) and GAL (Lin et al., 2019), our method offers higher accuracy with lower computational cost and fewer parameters.

### 7.8.2 VISUALIZING FEATURE PRESERVATION

We qualitatively evaluate the feature retention of the proposed pruning method on ResNet-50 using the ImageNet dataset. We randomly selected three images from ImageNet and compared the features extracted by the first five convolutional layers of the pruned network (with $r = 0.2$) to those of the original network. For each image, the top row displays the features of the original network, while the bottom row shows the features of the pruned network. As observed in Figure 3, there is no significant difference between the features extracted in these two scenarios.

Table 6: Compression results of ResNet-56 on CIFAR-10.

| Model | Parameters | FLOPs | Top1 (%) |
|---|---|---|---|
| Baseline (He et al., 2016) | 0.85M | 125.49M(0.0%) | 93.26 |
| GAL-0.6 (Lin et al., 2019) | 0.75M (11.8%) | 78.30M (37.6%) | 92.98 |
| L1 (;and Hans Peter Graf, 2017) | 0.73M (14.1%) | 90.90M (27.6%) | 93.06 |
| **Ours** | **0.71M (16.8%)** | 104.15M (17%) | **94.09** |
| HRank (Lin et al., 2020) | 0.71M(16.8%) | 88.72M(29.3%) | 93.52 |
| FilterSketch (Lin et al., 2021) | 0.68M(20.6%) | 88.05M(30.43%) | 93.65 |
| DECORE (Lin et al., 2019) | 0.64M (24.2%) | 92.48M(26.3%) | 93.34 |
| **Ours** | **0.59M (30.43%)** | 86.85M(30.78%) | **93.84** |
| SOKS (Liu et al., 2022) | 0.51M (40%) | 81.40M (36%) | 93.22 |
| FilterSketch (Lin et al., 2021) | 0.50M (41.2%) | 73.36M (41.5%) | 93.19 |
| NISP (Yu et al., 2018) | 0.49M (42.4%) | 81.00M (35.5%) | 93.01 |
| HRank (Lin et al., 2020) | 0.49M (42.4%) | 62.72M (50.0%) | 93.17 |
| HTP-URC (Qian et al., 2023) | 0.47M (44.72%) | 58.65M (46.58%) | 93.55 |
| DECORE (Lin et al., 2019) | 0.43M (49%) | 62.93M (49.9%) | 93.26 |
| **Ours** | **0.42M (49.97%)** | 62.12M (50.5%) | **93.54** |
| GAL-0.8 (Lin et al., 2019) | 0.29M (65.9%) | 49.99M (60.2%) | 90.36 |
| HRank (Lin et al., 2020) | 0.27M (68.1%) | 32.52M (71.69%) | 90.72 |
| QSFM (Wang et al., 2022) | 0.25M (71%) | 50.62M (60%) | 91.88 |
| CHIP (Sui et al., 2021) | 0.24M (71.8%) | 34.79M (72.3%) | 92.05 |
| FilterSketch (Lin et al., 2021) | 0.24M (71.8%) | 32.47M (74.4%) | 91.2 |
| **Ours** | **0.22M (74.12%)** | 32.18M (74.36%) | **92.56** |
| APIB (Guo et al., 2023) | 0.14M (83%) | 23.85M (81%) | **91.53** |
| **Ours** | **0.136M (84%)** | 20.08M (84%) | 91.39 |

Table 7: Pruning results for ResNet-110 on CIFAR-10.

| Model | Parameters | FLOPs | Top1 (%) |
|---|---|---|---|
| Baseline (He et al., 2016) | 1.7M(0.0%) | 252.89M(0.0%) | 93.50 |
| L1 (;and Hans Peter Graf, 2017) | 1.16M (32.6%) | 155M (38.7 %) | 93.3 |
| DECORE (Alwani et al., 2021) | 1.11M (36%) | 163.30M (35%) | 93.88 |
| HRank (Lin et al., 2020) | 1.04M(39.4%) | 148.70M(41.2%) | 94.23 |
| Ours | 0.97M (43.3%) | 142.804M (43.3%) | 94.2 |
| GAL-0.5 (Lin et al., 2019) | 0.95M(44.8%) | 130.20M(48.5%) | 92.55 |
| HRank (Lin et al., 2020) | 0.70M(59.2%) | 105.70M(58.2%) | 93.36 |
| **Ours** | **0.69M**(59.7%) | 101.09M(60.0%) | **93.91** |
| HRank (Lin et al., 2020) | 0.53M(68.7%) | 79.30M(68.6%) | 92.65 |
| **Ours** | **0.53M** (69.2%) | 77.284M (69.4%) | **93.28** |

## 7.9 EXPERIMENTAL SETUP

The pruning steps for our proposed method are as follows:

1. *Filter Selection:* The pruning ratio $r$ is uniformly applied across all convolutional layers, except the last one. Using Algorithm 1, the filters to be pruned are identified based on their importance scores.

2. *Pruning:* In a single step, the least important filters, identified in the previous phase, are pruned from all convolutional layers, excluding the final layer.

3. *Fine-Tuning:* To restore the network's accuracy post-pruning, the pruned model is fine-tuned.

Our code uses Pytorch 1.11.0. Our implementation is adapted from the code provided in (He et al., 2019), available Here, for both ResNet-50 on ImageNet and ResNet-56 on CIFAR-10. The dataset are the open-source ImageNet ( Here) and CIFAR-10 ( Here) datasets. The used pre-trained ResNet-50 on ImageNet is the pretarined model provided by PyTorch and the for ResNet-56 on CIFAR-10 we used the pre-trained model provided by (He et al., 2019), available Here. We are using PyCharm IDE for all the implemented codes. Pruning and fine-tuning was done in a combination of CPUs and GPU in both filter selection and fine-tuning. The characteristics of the used system is as follows

**System Configuration:**
Operating System: Windows 10 (Version 10.0.18363, Release 10)
CPU: Intel64 Family 6 Model 158 Stepping 13, GenuineIntel
Physical Cores: 16
Logical Cores: 16
Total RAM: 64 GB

**Python Configuration:**
Python Version: 3.9.12

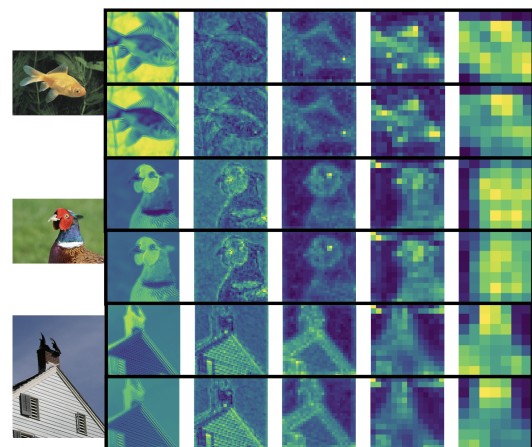

Figure 3: Style transfer results using ResNet-50 on three randomly selected images from the ImageNet dataset. For each image, the top row shows the output results from the first five convolutional layers of the original ResNet-50, while the bottom row displays the corresponding results from the pruned network using our proposed method with pruning ratio $r = 0.2$.

Python Compiler: MSC v.1916 64-bit (AMD64)

**PyTorch Configuration:**
PyTorch Version: 1.11.0
CUDA Version: 11.3 (cuDNN 8200)
GPU: 1 × NVIDIA GeForce RTX 2080 Ti with 11 GB GDDR6 memory

The fine-tuning setup details for results are as follows

**Fine-tuning setup for ResNet-50 on Imagenet:**

- Training Batch size: 64
- Testing Batch size: 32
- Epochs for fine-tuning: 110
- momentum = 0.9
- Optimizer for fine-tuning: Adam
- weight decay=0.0001
- Number of workers: 4
- Initial learning rate: e-03
- Learning rate schedule: Every 30 epochs
- Pruning rate: {0.2, 0.32, 0.72}

**Fine-tuning setup for ResNet-56 on CIFAR-10:**

- Training Batch size: 64
- Testing Batch size: 32
- Epochs for fine-tuning: varying between 200 and 300
- momentum = 0.9
- Optimizer for fine-tuning: Adam
- weight decay= 0.0005
- Number of workers: 2
- Initial learning rate: e-03

- Learning rate schedule: 3 times depending on the number of epochs.
- Pruning rate: $\{0.09, 0.17, 0.25, 0.50, 0.61\}$

