# OpenReview forum: "Green Pruning: Layer Interdependence-Aware CNN Pruning for Resource Efficiency"
_ICLR.cc/2026/Conference — ICLR 2026 Conference Withdrawn Submission_

### Official Review · Reviewer_jBNX · 2025-10-31

**Soundness:** 2
**Presentation:** 3
**Contribution:** 2
**Rating:** 2
**Confidence:** 4

**Summary:**

This paper argues that we should also reduce the cost of the pruning method itself for CNN pruning. To this end, the authors propose a resource efficiency metric and a data-free green pruning technique.

**Strengths:**

- It is interesting to see some discussions about the cost of the pruning method.

**Weaknesses:**

- Only focusing on the cost of the pruning method is not enough; the pruning method only needs to be conducted for once, but the pruned model will run for thousands or even millions of times.
- More recent baselines are needed for a direct comparison.

**Questions:**

- For Eq.(5) and Eq.(6), we see that the proposed method only focuses on the cost of the pruning method. In most cases, the pruning method only needs to be conducted once. However, after the pruned model is deployed in an application/production scenario, it will be run for thousands or even millions of times. For example, the current LLMs API, millions of people will call them around the world. The pruning cost is like $1\times Cost_{prune}$, but the cost for the deployed model is like $N_{calls}\times Cost_{model}$. In  most cases, $N_{calls} >> 1$. When designing a resource efficiency metric for green pruning, the reviewer thinks we should also consider this. The current metric is not comprehensive.

- Following the above point, in Table 2, the reviewer thinks the authors should also tell us the absolute value of the cost of pruning and the FLOPs of the model after pruning, not only the RE ratio. It would give us a better understanding of the pruning cost and the FLOPs of the pruned model. It is hard to get something from the RE ratio; the ACC and MAC are at different orders of magnitude, making the RE ratio something hard to understand.

- In Table 3, the reviewer finds that the most recent method is from the year 2021. Are there any more recent works? For example, RED++ and IPBM in Table 2, which come from the year 2023. Considering this is ICLR 2026, could the authors add some reference to methods in the years 2024 and 2025 for a straight comparison? This will better help us understand the effectiveness of the proposed method.

---

### Official Review · Reviewer_8fi7 · 2025-10-31

**Soundness:** 2
**Presentation:** 1
**Contribution:** 2
**Rating:** 2
**Confidence:** 2

**Summary:**

Response:This paper addresses two critical challenges in CNN pruning: high environmental cost and poor feasibility on edge devices. Its core contributions are twofold: 1) Proposing novel evaluation metrics—Resource Efficiency (RE), which quantifies the trade-off between pruned network accuracy and pruning computational cost, and Relative Carbon Efficiency (RCE), a system-agnostic metric enabling fair comparison of pruning methods’ sustainability by using FLOPs as a hardware-independent proxy for carbon footprint. 2) Developing a data-free "green pruning" method that explicitly models inter-layer dependencies to improve filter selection reliability and adopts a low-complexity oblivious algorithm leveraging γ-weak submodularity, avoiding iterative dataset passes. Experiments on ResNet-50, ResNet-56, and ResNet-110 claim to match SOTA accuracy while reducing computational overhead and carbon footprint by orders of magnitude compared to baselines like APIB, ThiNet, and HRank.

**Strengths:**

1. The inclusion of theorems (e.g., upper bound on output error in Theorem 4.1, weak submodularity in Theorem 4.2) provides a formal basis for the pruning method, distinguishing it from heuristic-only approaches.

2. Unlike prior work that links carbon footprint evaluation to specific hardware (hindering cross-method comparisons), the paper’s RCE metric is system-agnostic. It uses FLOPs (a hardware-independent proxy for runtime) to enable fair pruning method comparisons across setups without re-implementation on the same hardware.

**Weaknesses:**

1. The paper attributes improved performance to inter-layer dependency modeling but does not test a "no inter-layer" variant (e.g., using only current-layer norms). Without this, it is impossible to confirm if layer interdependence adds value beyond the oblivious algorithm or norm-based selection.

2. The paper includes post-pruning fine-tuning but does not account for its energy/carbon cost in RCE. Fine-tuning can dominate overall sustainability, so excluding it makes RCE an incomplete measure of real-world impact.

3. Experiments are limited to ResNet variants. Larger models (e.g., ViT) may expose scalability issues (e.g., increased complexity in layer dependency calculations) that the paper does not address.

4. This paper didn't provide any background or preliminaries of submodularity, which makes the theoretical parts in the main paper hard to follow.

5. Some typos:

* "Accoriding" and "equation equation 5" in line 143;
* In (8), it should be $||F||$, not $|F|$;
* ?? in line 866;
* (?) in line 934.

**Questions:**

1. In line 152: Why do you need to deploy a pruning method on the devices? Isn't it more common to prune the trained model on the server/cloud side and deploy the pruned model on the devices?

2. In (8), what's the insight for the design of this important function? In particular, why is it defined as $\max$ of the two norms, not multiplication or any other operation?

3. How does your method scale to larger architectures (e.g., ViT)? Do layer dependency calculations become computationally prohibitive for these models?

---

### Official Review · Reviewer_ujYe · 2025-10-31

**Soundness:** 1
**Presentation:** 2
**Contribution:** 1
**Rating:** 0
**Confidence:** 4

**Summary:**

This paper introduces a resource-efficient and environmentally-conscious pruning method for CNNs that explicitly models inter-layer dependencies. The authors propose two novel evaluation metrics: Resource Efficiency (RE) and Relative Carbon Efficiency (RCE) to assess pruning methods beyond traditional accuracy-FLOPs trade-offs. The core technical contribution is a γ-weakly submodular importance function based on filter norms across adjacent layers, solved using a low-complexity oblivious algorithm. Experiments on ResNet architectures demonstrate competitive accuracy with substantially reduced computational overhead during the pruning process itself.

**Strengths:**

1. RE and RCE metrics address an important gap in pruning literature by considering sustainability

2. Green AI and sustainable ML are increasingly important

**Weaknesses:**

1. The paper claims exceptional computational efficiency compared to APIB (Guo et al., 2023), which is an automatic pruning method. However, there exist many other **norm-based pruning methods** that can also obtain filter rankings very quickly without requiring greedy search. For instance, simple L1-norm pruning or L2-norm pruning can rank all filters in a single pass by computing norms independently for each filter, potentially achieving even faster computation than the proposed method. These approaches similarly avoid iterative dataset passes and greedy algorithms. Therefore, it is unclear whether the added complexity of considering inter-layer dependencies provides sufficient benefit to justify the method's claim.

2. Incomplete carbon footprint analysis. While RCE is system-agnostic, the paper doesn't validate actual energy consumption measurements. The FLOP-to-energy conversion assumes a constant relationship, which may not hold across different operations or hardware states.

3. Based on point 1 and 2, I believe there is a huge gap between the title and actual contribution.

4. The compared methods are old. From line 412- 416, "HRank (Lin et al., 2020), RED++ (Yvinec et al., 2023), ThiNet (Luo et al., 2018), GAL (Lin et al., 2019), SSS (Huang and Wang, 2018), L1 (;and Hans Peter Graf, 2017), F-ThiNet (Tofigh et al., 2022), He et al., (He et al., 2017), GDP (Lin et al., 2018), NISP (Yu et al., 2018), DECORE (Alwani et al., 2021), FilterSketch (Lin et al., 2021), and APIB (Guo et al., 2023)." None of these is recent. The authors should cite and compare papers in the year 2024 and 2025.

**Questions:**

See weakness

---

### Official Review · Reviewer_6uSH · 2025-11-02

**Soundness:** 2
**Presentation:** 2
**Contribution:** 2
**Rating:** 2
**Confidence:** 4

**Summary:**

# GREEN PRUNING

## Summary

While studying pruning methods, we typically measure the reduction in inference-time computation (FLOPs) due to pruning. Instead, this work measures the computation cost of performing the pruning itself — that is, the computation cost of selecting the filters for pruning. Furthermore, it measures the cost of pruning in terms of carbon footprint, using a system-agnostic framework.

Two measures are proposed: **Runtime Carbon Efficiency (RCE)** and **Resource Efficiency (RE)**.
- **RCE** is defined as the ratio between the number of operations required by two pruning methods.
- **RE** is defined as the ratio between accuracy and the number of multiply–accumulate (MAC) operations required in the pruning process.

Finally, a pruning method termed the *Oblivious Algorithm* has been proposed.

**Strengths:**

## Strengths

- The paper delves into an important area — the pruning efficiency of pruning methods themselves.
- The two proposed measures, **RCE** and **RE**, seem easy to implement and offer a system-agnostic strategy for evaluating pruning methods.
- Tables 1, 2, and 3 present a convincing case that the proposed solution is superior compared to other baselines.

**Weaknesses:**

## Comments

- It is not clear how RCE and RE result in a reduction of the carbon footprint. While some relationship is established in Equations (1)–(5), there are no results that demonstrate the actual amount of reduction in carbon footprint.

- The claim of a system-agnostic framework is also not evaluated. An analysis showing the true carbon footprint of a particular pruning method across multiple hardware platforms and its comparison with the proposed strategy is needed to better understand the effectiveness of the proposed metrics and algorithm.

- The Oblivious Algorithm, though offering low complexity due to its simplified nature, presents results in Tables 1, 2, and 3 that convincingly demonstrate the superiority of the proposed solution compared to other baselines. However, since it is based on the L₂-norm of filters, it is unclear how it achieves lower computational cost or eventually a lower carbon footprint.

- Furthermore, the computations depend on the number of epochs required by each method to achieve the reported results, but there is no discussion of computation or carbon footprint per epoch.

- While the method is compared with SOAT approaches, a more categorical comparison such as with Knowledge Distillation, Structured online/offline pruning, recent methods for transformers could be useful in also help in evaluating whether ti can be extended to more recent models such as those based on transformers.

## Minor Comments

- In Equation (3), it is not clear what PUE refers to.
- The paper is 9 pages long, whereas the Appendix extends over 12 pages, which is excessive. Only relevant information should be included in the Appendix, and more important information should be incorporated into the main paper.

**Questions:**

Please refer to Weaknesses section

---

### Note · Authors · 2025-11-12

I have read and agree with the venue's withdrawal policy on behalf of myself and my co-authors.